# A novel SARS-CoV-2 related coronavirus in bats from Cambodia

Deborah Delaune [1,2,3,14], Vibol Hul [4,5,14], Erik A. Karlsson [4,14], Alexandre Hassanin [6], Tey Putita Ou [4], Artem Baidaliuk [1], Fabiana Gámbaro[1,7], Matthieu Prot[1], Vuong Tan Tu[6,13], Sokha Chea[8], Lucy Keatts[9,10], Jonna Mazet[10], Christine K. Johnson[10], Philippe Buchy [4,11], Philippe Dussart [4,12], Tracey Goldstein [10], Etienne Simon-Lorière [1,15✉] & Veasna Duong [4,15✉]

Knowledge of the origin and reservoir of the coronavirus responsible for the ongoing COVID-19 pandemic is still fragmentary. To date, the closest relatives to SARS-CoV-2 have been detected in *Rhinolophus* bats sampled in the Yunnan province, China. Here we describe the identification of SARS-CoV-2 related coronaviruses in two *Rhinolophus shameli* bats sampled in Cambodia in 2010. Metagenomic sequencing identifies nearly identical viruses sharing 92.6% nucleotide identity with SARS-CoV-2. Most genomic regions are closely related to SARS-CoV-2, with the exception of a region of the spike, which is not compatible with human ACE2-mediated entry. The discovery of these viruses in a bat species not found in China indicates that SARS-CoV-2 related viruses have a much wider geographic distribution than previously reported, and suggests that Southeast Asia represents a key area to consider for future surveillance for coronaviruses.

[1] Evolutionary Genomics of RNA Viruses, Department of Virology, Institut Pasteur, Paris, France. [2] Institut de Recherche Biomédicale des Armées, Brétigny-sur-Orge, France. [3] Université Paris-Saclay, Orsay, France. [4] Virology Unit, Institut Pasteur du Cambodge, Institut Pasteur International Network, Phnom Penh, Cambodia. [5] UVE: Aix-Marseille Univ-IRD 190-Inserm, 1207 Marseille, France. [6] Institut de Systématique, Évolution, Biodiversité, Sorbonne Université, MNHN, CNRS, EPHE, UA, Paris, France. [7] Université de Paris, Sorbonne Paris Cité, Paris, France. [8] Wildlife Conservation Society, Cambodia Program, Phnom Penh, Cambodia. [9] Wildlife Conservation Society, Health Program, Bronx, NY, USA. [10] One Health Institute, School of Veterinary Medicine, University of California, Davis, USA. [11] GlaxoSmithKline Vaccines R&D Greater China & Intercontinental, Singapore, Singapore. [12] Virology Unit, Institut Pasteur de Madagascar, Institut Pasteur International Network, Antananarivo, Madagascar. [13] Present address: Institute of Ecology and Biological Resources, Vietnam Academy of Science and Technology, Hanoi, Vietnam. [14] These authors contributed equally: Deborah Delaune, Vibol Hul, Erik A. Karlsson. [15] These authors jointly supervised this work: Etienne Simon-Lorière, Veasna Duong. ✉email: etienne.simon-loriere@pasteur.fr; dveasna@pasteur-kh.org

O ver a year has passed since the emergence of Severe Acute Respiratory Syndrome coronavirus 2 (SARS-CoV-2)[1], responsible for the ongoing coronavirus disease 2019 (COVID-19) pandemic. However, information on the origin, reservoir, diversity, and extent of circulation of ancestors to SARS-CoV-2 remains scarce. Horseshoe bats (genus *Rhinolophus*) are believed to be the main natural reservoir of SARS-related coronaviruses also named Sarbecoviruses[2]. Indeed, a high diversity of coronavirus species have been found in *Rhinolophus* bats collected in several provinces of China[3]. To date, the closest relatives to SARS-CoV-2 were identified from horseshoe bats sampled in the Yunnan province, southern China[1,4,5]. RaTG13 was sequenced from a *Rhinolophus affinis* bat in 2013, RmYN02 from a *Rhinolophus malayanus* bat in 2019, and RpYN06 from a *Rhinolophus pusillus* in 2020. Two viruses were also detected in Sunda pangolins (*Manis javanica*) seized in two provinces of southern China[6]. More distant and highly mosaic recombinant viruses were also sampled from bats in the Zhejiang province, in eastern China in 2015 and 2017[7]. Southeast Asia is considered a hotspot for emerging diseases[8]. More than 25% of the world's bat diversity is found there[9], and a close relative of SARS-CoV-2 was identified in bats captured in a cave in Thailand in June 2020[10]. In this work we report the identification and characterization of two coronaviruses closely related to SARS-CoV-2 in bats sampled in Cambodia in 2010, indicating that this viral lineage circulates in a much wider geographic area than previous reported.

## Results

**Testing of archived samples**. Following the emergence of COVID-19, to search for putative SARS-CoV-2-like betacoronaviruses (betaCoVs) in Cambodia, 430 archived samples from six bat families and two carnivoran mammal families, including 162 oral swabs and 268 rectal swabs, were retrospectively tested with a pan-coronavirus (pan-CoV) hemi-nested RT-PCR[11] (Supplementary Table 1). Sixteen rectal swabs out of 430 (3.72%) samples tested positive for CoV by pan-CoV hemi-nested PCR. Eleven were classified as alphacoronaviruses and five as betaCoV. Two of the five betaCoV samples further tested positive using a RT-qPCR targeting the RdRp gene of sarbecoviruses[12]. Both samples came from rectal swabs of Shamel's horseshoe bats (*Rhinolophus shameli*) sampled in December 2010 in the Steung Treng province in Cambodia. Oral swabs from these same *R. shameli* bats tested negative for the presence of betaCoV RNA, despite the high proportion of reads matching the coronavirus (23%) in the rectal swab of RshSTT200.

**Phylogenetic characterization of RshSTT182 and RshSTT200**. RNA samples were then processed for next-generation metagenomic sequencing, using a ribosomal RNA depletion approach and randomly primed cDNA synthesis[13]. Reads assembly reconstructed two nearly identical coronavirus genomes, named BetaCoV/ Cambodia/RshSTT182/2010 (RshSTT182) and BetaCoV/Cambodia/RshSTT200/2010 (RshSTT200), respectively. The two sequences are closely related to SARS-CoV-2, exhibiting 92.6% nucleotide identity across the genome (Supplementary Table 2) and identical genomic organization. Phylogenetic analysis using full genome sequences shows that RshSTT182 and RshSTT200 represent a sublineage of SARS-CoV-2 related viruses, despite the geographic distance of isolation (Fig. 1). Genetic similarity with SARS-CoV-2 is maintained across the genome, with the exception of a portion corresponding to the spike N terminal domain (NTD; Fig. 2 and Supplementary Fig. 1). In several sections of the genome, including the region spanning

nsp4 to nsp8 within orf1a, RshSTT182, and RshSTT200 are genetically closer to SARS-CoV-2 than any other closely related viruses discovered to date. Similarity is further evidenced when inferring phylogeny based on the sequence coding for these proteins.

Extensive evidence exists on numerous recombination events in the evolutionary history of the sarbecoviruses[14–16]. Consistent with this, we found that both RshSTT182 and RshSTT200 are also mosaic viruses (Fig. 2 and Supplementary Fig. 1); however, most regions identified as recombinant in origin appear to have involved close relatives within the SARS-CoV-2 sublineage. Only a region encompassing the Spike N terminal domain (NTD) is closer to more distantly related betaCoVs. In all other regions of the genome, the viruses detected in Cambodia consistently branch as a sister clade to SARS-CoV-2 and RaTG13, with minor swaps in the subtree topology. Interestingly, both regions showing high similarity to SARS-CoV-2 (nsp4 to 8 within orf1a and orf8) overlap with regions identified as recombinant. All these elements suggest a co-circulation of ancestors to these viral sublineages with both a wider geographic area and more distinct bat species than those previously identified. Of note, the current geographic distribution of *R. shameli* bats does not include China (Supplementary Figs. 2 and 3)[17]. However, the distributions of *R. affinis, R. pusillus,* and *R. malayanus* overlap with *R. shameli* distribution area in Southeast Asia, and extend into China, including the Yunnan province where the other viruses closely related to SARS-CoV-2 were detected. *R. affinis* and *R. malayanus* bats were concomitantly captured in the same northern karst region where these *R. shameli* bats were sampled in 2010, and transmission of coronaviruses is common among *Rhinolophus* species, especially when co-roosting in the same cave[18,19]. Finally, the haplotype network of *R. shameli* CO1 sequences shows a typical star-like pattern, suggesting that populations of *R. shameli* found between northern Cambodia and northern Laos are not genetically isolated[20].

**Analysis of RshSTT200 receptor binding domain and function**. Further risk assessment is needed to understand the host range (including humans) and pathogenesis associated with this SARS-CoV-2 sublineage. Homology modeling suggests that the external subdomain of the spike receptor binding domain (RBD) structure is highly similar to SARS-CoV-2 (Fig. 3a). We note the shortening of a loop at the beginning of the receptor binding motif and the presence of a conserved disulfide bond. Interestingly, five of the six amino acid residues reported to be major determinants of efficient receptor binding of SARS-CoV-2 to the human angiotensin-converting enzyme 2 (hACE2) receptor[21] are conserved. However, pseudoviral particles expressing the RshSTT200 spike were not able to infect HEK293T cells expressing hACE2 (Fig. 3c), while they were able to infect HEK293T expressing *R. shameli* ACE2 (RshACE2). The HEK293T cells expressing RshACE2 also allowed entry of pseudoviral particles expressing the SARS-CoV-2 spike (Fig. 3d) although to a lesser extent than hACE2, and in accordance with its reported wide tropism[22]. Finally, the poly-basic (furin) site present in SARS-CoV-2 is absent in both RshSTT182 and RshSTT200.

## Discussion

The data presented here further indicate that SARS-CoV-2 related viruses have a much wider geographic distribution than previously understood, and likely circulate via multiple *Rhinolophus* species. Our current understanding of the geographic distribution of the SARS-CoV and SARS-CoV-2

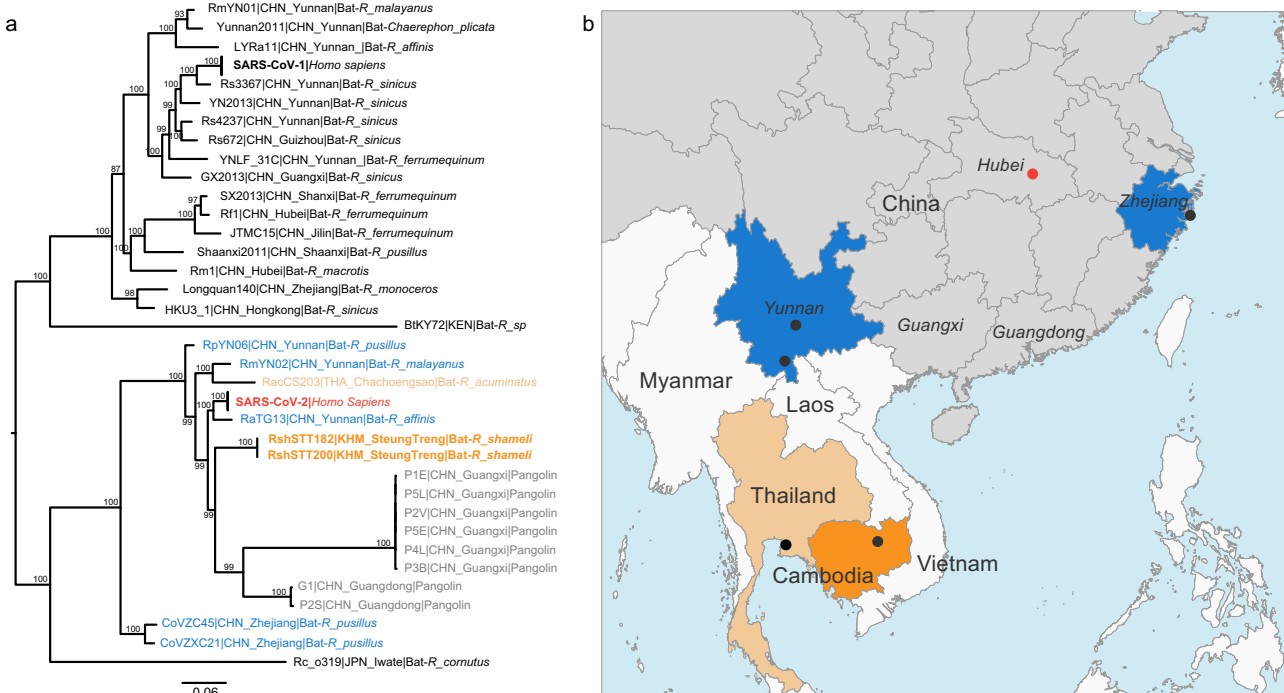

**Fig. 1 Phylogenetic analysis of SARS-CoV-2 and representative sarbecoviruses and geographical context. a** Maximum likelihood phylogeny of the subgenus *Sarbecovirus* (genus *Betacoronavirus*; n = 39) estimated from complete genome sequences using IQ-TREE and 1000 replicates. The coronaviruses of the SARS-CoV-2 lineage are color coded by country of sampling as on the map. In orange, Cambodia, light orange, Thailand and blue, China. Taxa names include the isolate name, country and province of sampling, and host. The scientific names of the hosts are abbreviated as follows: Bats: *R. affinis*, *Rhinolophus affinis*; *R. sinicus*, *Rhinolophus sinicus*; *R. ferrumequinum*, *Rhinolophus ferrumequinum*; *R. malayanus*, *Rhinolophus malayanus*; *R. acuminatus*, *Rhinolophus acuminatus*; *C. plicata*, *Chaerephon plicata*; *R. pusillus*, *Rhinolophus pusillus*; *R. macrotis*, *Rhinolophus macrotis*; *R. monoceros*, *Rhinolophus monoceros*; *R. cornutus*, *Rhinolophus cornutus*; Pangolin: *M_javanica*, *Manis javanica* and human: *H. sapiens*, *Homo sapiens*. A maximum clade credibility tree is available in Supplementary Fig. 3. **b** map of parts of China and Southeast Asia. Regions where viruses of the SARS-CoV-2 lineage were sampled are colored as in the tree. A black dot indicates a sampling site when known, and the red dot shows the location of Wuhan, where the first cases of SARS-CoV-2 infection were reported.

lineages[14] possibly reflects a lack of sampling in Southeast Asia, or at least across the Greater Mekong Subregion, which encompasses Myanmar, Laos, Thailand, Cambodia and Vietnam, as well as the Yunnan and Guanxi provinces of China, linking the sampling area of the closest viruses to SARS-CoV-2 identified to date. Finally, pangolins, as well as members of order *Carnivora*, especially the *Viverridae*[5], *Mustelidae*[6], and *Felidae*[7] families are readily susceptible to SARS-CoV-2 infection, might represent intermediary hosts for transmission to humans, and should not be ignored in future surveillance efforts in the region. Viruses of the SARS-CoV-2 sublineage, with one exhibiting strong sequence similarity to SARS-CoV-2 in the RBD, were recently detected in distinct groups of pangolins seized during anti-smuggling operations in southeast China[6]. While it is not possible to know where these animals became infected, it is important to note that the natural geographic range of the pangolin species involved (*Manis javanica*) also corresponds to Southeast Asia and not China.

Southeast Asia, which hosts a high diversity of wildlife and where exists extensive trade in and human contact with wild hosts of SARS-like coronaviruses, may represent an area to consider in the ongoing search for the origins of SARS-CoV-2[23], and certainly in broader coronavirus surveillance efforts. The region is undergoing dramatic land-use changes such as infrastructure development, urban development, and agricultural expansion, that can increase contacts between bats, other wildlife, and humans. Continued and expanded surveillance of bats and other key wild animals in Southeast Asia is thus a crucial component of future pandemic preparedness and prevention.

## Methods

**Ethics statement**. The study was approved by the General Directorate of Animal Health and Production and Forest Administration department of the Ministry of Agriculture Forestry and Fisheries in Cambodia. Sampling was conducted under a University of California, Davis Institutional Animal Care and Use Committee approved protocol (UC Davis IACUC Protocol No. 19300). The bat capture and sampling in 2010 was authorized by UNESCO and the National Authority of Preah Vihear.

**Sampling**. Testing was performed on archived samples from several programs and field missions (Supplementary Table 1). In 2010, the Muséum national d'Histoire naturelle (MNHN, Paris, France) was mandated by UNESCO and the National Authority of Preah Vihear to conduct a mammal survey in northern Cambodia. During this mission, bats were captured using mist nets and harp traps in two provinces, Preah Vihear and Ratanakiri, to compare bat diversity on the two sides of the Mekong River. One site of bat capture was later identified using GPS coordinates to in fact be a cave in Stung Treng province, close to the border of Preah Vihear province. Bats were morphologically identified at the species level by AH and VTT.

More recent sampling efforts were supported by the USAID-funded PREDICT project, which aimed to strengthen global capacity for detection and discovery of viruses with pandemic potential that can move between animals and people. From 2012 to 2018, samples from bats and carnivorans were collected from free-ranging animals, private animal collection, restaurant, or hunted animals in Battambang, Kampong Cham, Mondulkiri, Preah Vihear, Pursat, Ratanakiri, and Stung Treng. Mist nets were used to catch free-ranging bats. Oral and rectal swabs were collected from live animals which were released immediately after sampling.

The samples from these sampling missions were stored in viral transport medium solution containing tryptose phosphate broth 2.95%, 145 mM NaCl, 5% gelatin, 54 mM amphotericin B, 106 U penicillin-streptomycin per liter, 80 mg gentamycin per liter (Sigma-Aldrich) and were held in liquid nitrogen in dewars for transport to the Institut Pasteur du Cambodge where they were stored at −80 °C prior to testing.

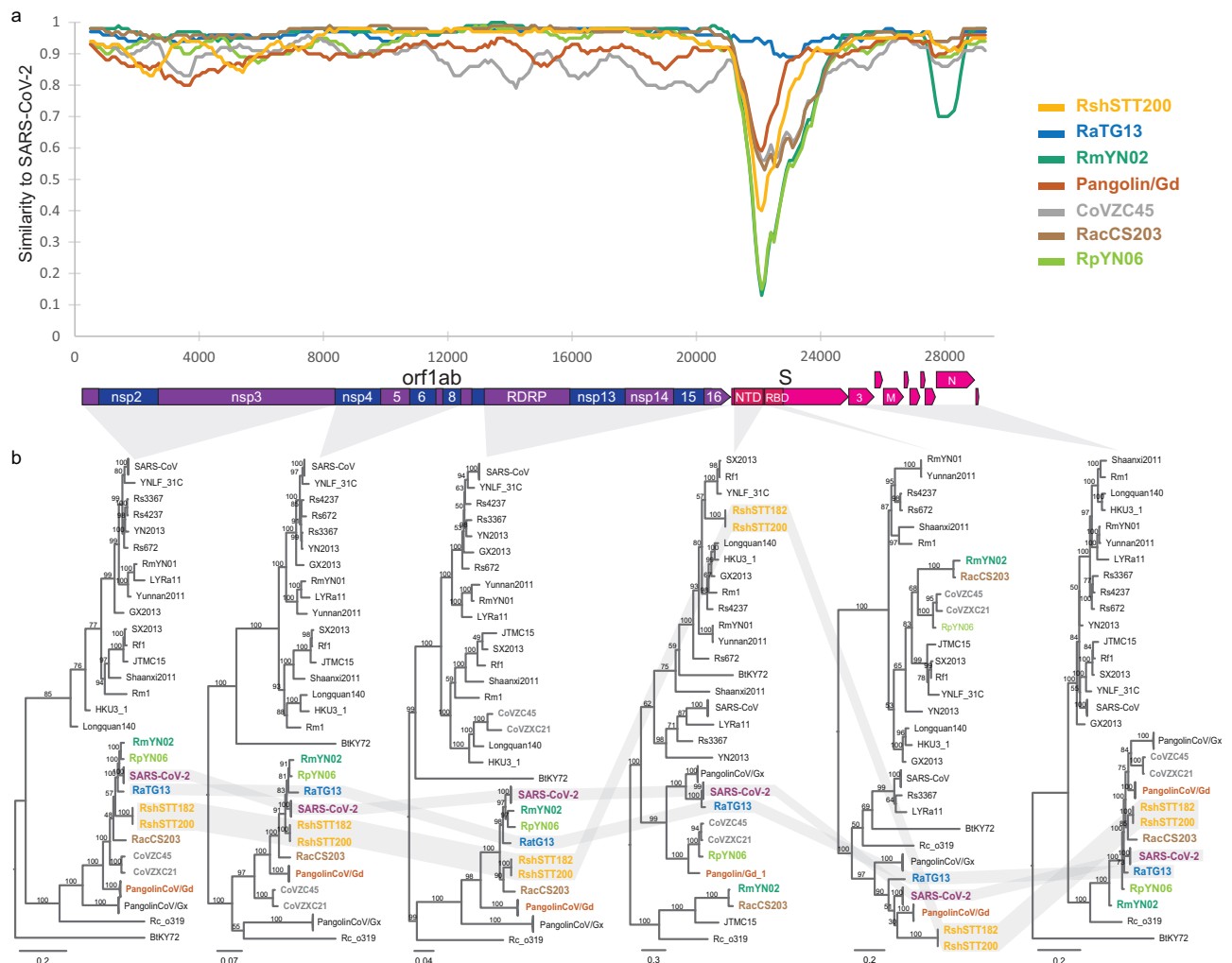

**Fig. 2 Recombination analysis. a** Sliding window analysis of changing patterns of sequence similarity between SARS-CoV-2 and related coronaviruses from China and Cambodia. CoVZXC21 and CoVZC45 were merged for this analysis. Source data are provided as a Source Data file. **b** Phylogenetic tree of different genomic regions. From left to right, region spanning: nsp1-nsp3, nsp4-nsp8, rdrp-nsp16, spike N terminal domain (NTD), spike receptor binding domain (RBD), orf3-N. Branch support obtained from 1000 bootstrap replicates are shown. SARS-CoV and SARS-CoV-2 sequences are collapsed, and trees are midpoint rooted for clarity.

The samples were selected and tested for SARS-CoV-2 related virus through an effort to look at previously-collected samples that were not initially prioritized for testing nor been tested with RT-PCR assays capable of detecting SARS-CoV-2 related viruses due to resource constraints.

The two bats positive for viruses closely related to SARS-CoV-2 were collected during the MNHN mission, and were morphologically identified as *Rhinolophus shameli*, with their taxonomic status were further confirmed by analyzing the sequences of the *cytb* gene and the subunit 1 of the *cytochrome c oxidase* gene (CO1) (Supplementary Fig. 3).

**RNA extraction and qRT-PCR**. RNA from rectal swabs was extracted using QIAamp® Viral RNA kits (Qiagen). The samples were tested with a pan-coronavirus (pan-CoV) hemi-nested RT-PCR[11] and by a RT-qPCR known to detect sarbecoviruses[12], including SARS-CoV-2. A large fraction of these samples has been previously tested with another pan-CoV RT-PCR[24], which does not detect SARS-CoV-2 like viruses. Initial viral isolation attempts were unsuccessful but further isolation is being attempted in several bat cell lines.

**Next generation sequencing**. Extracted RNA was treated with Turbo DNase (Ambion) followed by purification using SPRI beads (Agencourt RNA clean XP, Beckman Coulter). We used a ribosomal RNA (rRNA) depletion approach based on RNAse H and targeting human rRNA[13]. The RNA from the selective depletion was used for cDNA synthesis using SuperScript IV (Invitrogen) and random primers, followed by second-strand synthesis. Libraries were prepared using a Nextera XT kit (Illumina) and sequenced on an Illumina NextSeq500 (2 × 75 cycles).

**Genome assembly**. Raw reads were trimmed using Trimmomatic v0.39[25] to remove adapters and low-quality reads. We assembled reads using the metaspades option of SPAdes/3.14.0[26] and megahit v1.2.9[27] with default parameters. Scaffolds were queried against the NCBI non-redundant protein database[28] using DIA-MOND v2.0.4[29]. Among other putative viruses (hits summarized in Supplementary Table 3), the *Sarbecovirus* genomes identified were verified and corrected by iterative mapping using CLC Assembly Cell v5.1.0 (Qiagen). Aligned reads were manually inspected using Geneious prime v2020.1.2 (2020) (https://www.geneious.com/), and consensus sequences were generated using a minimum of 3× read-depth coverage to make a base call. The genomes are nearly identical, presenting three nucleotides difference between them: g12196a, c20040t, and t24572c). We used Ivar[30] to estimate the frequency of minor variants (iSNV) from the coronavirus reads. Coverage depth and iSNVs are reported in Supplementary Fig. 4. The sequence of the spike gene of each virus was confirmed by Sanger sequencing, using primers listed in Supplementary Table 4. The sequence of *Rhinolophus shameli* ACE2 gene was similarly reconstructed from the reads.

**Dataset**. Complete genome sequence data and metadata of representative SARS-like viruses were retrieved from GenBank, ViPR[31], and GISAID. Sequences were aligned by MAFTT v.7.467[32], and the alignment checked for accuracy using MEGA v7[33]. Accession numbers of all 39 sequences are available in Supplementary Table 5. Separate alignments were generated for the main ORFs.

The nucleotide similarities shown in SimPlot[34] analysis were generated by using a Kimura 2 parameter distance model with a 1000-nt sliding window moved along the sequence in 100-nt increments.

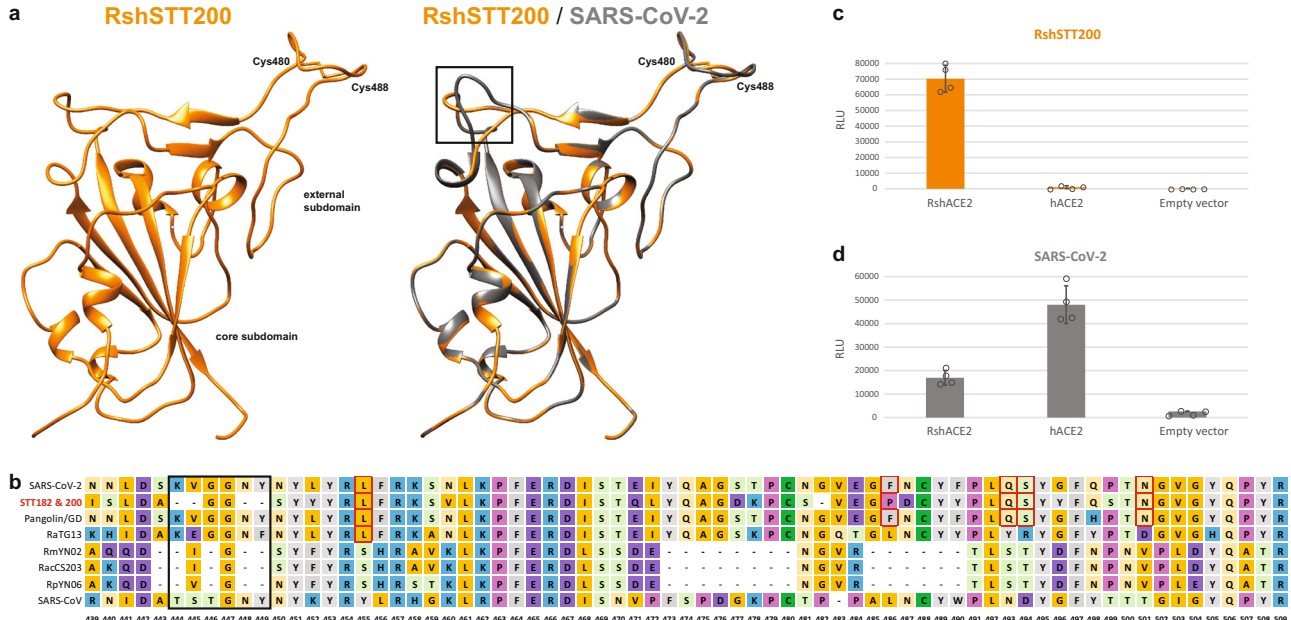

**Fig. 3 Structural modeling. a** Homology modeling of the RBD structure. The three-dimensional structure of the RshSTT200 Spike RBD was modeled using the Swiss-Model program employing the structure of SARS-CoV-2 (PDB: 6yla.1) as a template. The core and external subdomains are colored orange, and gray for RshSTT200 and SARS-CoV-2, respectively. The shortening a loop near the receptor-binding site of RshSTT200 are indicated by a black rectangle. The cysteines involved in a conserved disulfide bond are indicated. **b** Alignment of the receptor binding motif amino acid sequences of selected betaCoVs. **c** RshSTT200 spike (deleted for the last 21 amino acids) pseudovirus entry into HEK293T cells transfected with either RshACE2, hACE2 or an empty vector (pLenti-puro). **d** SARS-CoV-2 spike pseudovirus entry into HEK293T cells transfected with either RshACE2, hACE2 or an empty vector (pLenti-puro). Data are represented as mean ± standard deviation of technical replicates ($n = 4$) and are representative of three independent experiments. Source data are provided as a Source Data file.

**Recombination analysis.** We used a combination of six methods implemented in RDP5[35] (RDP, GENECONV, MaxChi, Bootscan, SisScan, and 3SEQ) to detect potential recombination events, and conservatively considered recombination signal detected by at least five methods. The beginning and end of breakpoints identified with RDP5 were used to split the genome into regions for further phylogenetic analysis.

**Phylogenetic analysis.** Maximum-likelihood (ML) phylogenies were inferred using IQ-TREE v2.0.6[36] and branch support was calculated using ultrafast bootstrap approximation with 1000 replicates[37]. Prior to the tree reconstruction, the ModelFinder application[38], as implemented in IQ-TREE, was used to select the best-fitting nucleotide substitution model. Bayesian phylogenies were also inferred using MrBayes v3.2.7[39], using the GTR substitution model. Ten million steps were run and parameters were sampled every 1000 steps.

**Structure modeling.** The three-dimensional structure of the RBD of RshSTT200 was modeled using the SWISS-MODEL program[40], using SARS-CoV-2 (PDB: 6yla.1) structure as it was the best hit for the RshSTT200 amino acid sequence input.

**Pseudovirus entry assay.** HEK293T cells (Sigma) were maintained in complete medium (DMEM, Gibco) with 10% fetal bovine serum (Gibco) and 1% penicillin-streptomycin (Gibco).

The sequence of *Rhinolophus shameli* ACE2 gene, codon-optimized for human expression, was synthetized (GeneArt, ThermoFischer) and cloned into an expression plasmid pLenti-puro-RshACE2. pLenti-puro was a gift from Ie-Ming Shih (pLenti-puro, Addgene #39481)[41]. The sequence corresponding to the spike gene of RshSTT200 deleted of the last 21 amino acids and codon-optimized for human expression was de novo synthesized (GeneArt,ThermoFischer) and cloned into the pHDM expression plasmid from the lentiviral kit. The sequence of each insert was verified by Sanger sequencing.

Lentivirus pseudoparticles packaging a coronavirus spike were produced using the system described by the Bloom laboratory[42]. The following reagent was obtained through BEI Resources, NIAID, NIH: SARS-Related Coronavirus 2, Wuhan-Hu-1 Spike-Pseudotyped Lentiviral Kit, NR-52948, kindly contributed by Alejandro B. Balazs and Jesse D. Bloom. Briefly, HEK293T were seeded in 10 cm dishes. The next day, the cells were co-transfected with 10 μg of pHAGE-CMV-Luc2-IRES-ZsGreen-W (NR-52516), 3.33 μg each of helper plasmids HDM-Hgpm2 (NR-52717), HDM-tat1b (NR-52518), and pRC-CMV-Rev1b (NR-52519), and 5 μg of a spike expressing plasmid expressing either the RshSTT200 spike or the complete SARS-CoV-2 spike (NR52514) with CaCl₂. Supernatants were collected 72 h of post-transfection, clarified by centrifugation, aliquoted and frozen at −80 °C.

To assay entry, HEK293T were seeded in 96-wells plates one day prior to transfection with pLenti-puro-RshACE2, pHAGE2-EF1aInt-ACE2-WT (NR52512) or pLenti-puro (empty) using Lipofectamine 3000 (Invitrogen) according to the manufacturer's protocol. The day after transfection, media was removed and cells were transduced with pseudoparticles expressing either spike with 5 μg/ml of polybrene transfection reagent (Merck-Millipore) in a final volume of 150 μl. Three days later, an equal volume of Bright Glo reagent (Promega) was added and mixed by pipetting. After 10 min of incubation, quantification was done with a Centro XS LB 960 (Berthold technologies). Three independent replicates were performed.

**Reporting summary.** Further information on research design is available in the Nature Research Reporting Summary linked to this article.

## Data availability

The data generated in this study have been deposited in the European Nucleotide Archive database under accession code PRJEB42502. The consensus sequences of RshSTT182 and RshSTT200 are also available at the GISAID[43] database with accession numbers: EPI_ISL_852604 and EPI_ISL_852605 [https://www.gisaid.org/]. The sequence of *Rhinolophus shameli* ACE2 gene has been deposited under GenBank accession number MZ851782. Source data are provided with this paper.

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

## Acknowledgements

We thank the government of Cambodia for permission to conduct this work. We thank also General Directorate of Animal Health and Production, Department of Wildlife and Biodiversity, Forestry Administration, Ministry of Agriculture, Forestry and Fisheries, Communicable Disease Control Department, Ministry of Health, the Wildlife Conservation Society teams and all students who helped collecting field samples. We extend our gratitude to the Virology Unit team at Institut Pasteur du Cambodge for technical support in laboratory diagnostic, and to Gabor Csorba for providing three *Rhinolophus shameli* samples. We are grateful to all researchers who have kindly shared genome data on the International Nucleotide Sequence Database Collaboration or on the GISAID. Supplementary Table 6 lists the originating and contributing laboratories of the sequences retrieved on the GISAID for this work. This study was made possible by the generous support of the American people through the United States Agency for International Development (USAID) Emerging Pandemic Threats PREDICT project (cooperative agreement number GHN-A-OO-09-00010-00 and AID-OAA-A-14-00102), with a specific extension for the testing reported here. V.H. is supported by a scholarship from the French Government (BGF) for his Ph.D. E.S.L. acknowledges funding from the French Government's Investissement d'Avenir program, 'INCEPTION' (ANR-16-CONV-0005), and Laboratoire d'Excellence 'Integrative Biology of Emerging Infectious Diseases' (ANR-10-LABX-62-IBEID). In 2010, the fieldwork was supported by the National Authority for Preah Vihear, UNESCO, "Société des amis du Muséum et du Jardin des Plantes", and the Muséum national d'Histoire naturelle.

## Author contributions

E.S.-L, P.D, T.G., and V.D. designed the research. E.S.-L and V.D. supervised the research. S.C., L.K., J.M., C.K.J., and P.B. provided resources. A.H., V.T.T., and V.H. collected bats samples. V.H. screened samples. D.D., F.G., and E.S.-L. performed the metagenomic sequencing. D.D., F.G., A.B., and E.S.-L. performed genome assembly and annotation. P.O.T. and E.S.L performed the structural modeling. D.D., A.B., P.O.T., V.D., and E.S.-L. performed the genome analysis and interpretation. D.D and M.P performed the pseudovirus entry assay. E.A.K. and E.S.-L. wrote the paper with inputs from all authors. All authors took part in data interpretation and edited the paper.

## Competing interests

P.B. is currently an employee of GSK vaccines. The remaining authors declare no competing interests.

## Additional information

**Peer review information** *Nature Communications* thanks Zheng-Li Shi and the other anonymous reviewer(s) for their contribution to the peer review this work. Peer reviewer reports are available.

