## [Peer Review File · Nature Communications]

Reviewers' Comments:

Reviewer #1:

Remarks to the Author:

In the present study, Hul and colleagues described a novel SARS-CoV-2 related coronavirus in bats from Cambodia. This report is important and interesting and reads well. A few concerns should be addressed.

1. Highlighting Guangxi and Guangdong in Figure 1b does not make sense because the pangolins were smuggled into the two regions from Southeast Asia.
2. A recent study described a SARS-CoV-2 related coronavirus from Thailand (<https://www.nature.com/articles/s41467-021-21240-1>). I suggest that the authors include this strain into their analysis.
3. The abstract section is quite descriptive and should describe more results about the two novel strains.
4. Interestingly, the two strains shared identical the same amino acid motifs critical for spike-ACE2 binding. It would be more convincing if the authors confirm the spike gene sequence using Sanger sequencing.
5. "than the any other closely related viruses discovered to date" Please remove 'the'.

Reviewer #2:

Remarks to the Author:

Hul et al., has retrospectively identified two SARS-CoV-2 related coronavirus genome sequences (RshSTT182 and RshSTT200) from *Rhinolophus shameli* bats sampled in Cambodia in 2010 which are genetically related to SARS-CoV-2. While the finding is not totally novel as similar viruses have been reported in bats from Japan, China to Thailand. The current findings further highlight that Southeast Asia is potentially a hotspot for SARS-CoV-2 related coronaviruses. The manuscript is concise and well-written. However, a number of points need to be addressed to improve the manuscript:

MAJOR POINTS

- 1) If it is possible at all, the authors should include serology data (using archived samples is ok) to further support the prevalence of the viruses in *Rhinolophus shameli* bats in Cambodia
- 2) For all of the molecular analyses, please repeat by including the recently published RacCS203 genome sequence (published by Wacharapluesadee et al in Nat Comm) from Thailand and the Sc-o319 from Japan
- 3) The ability of RBD to bind hACE2 is a hallmark of risk assessment for potential spillover of bat CoVs. The authors made some discussion/prediction in the paper, but purely based on protein sequence analysis only. As the biochemical analysis of RBD-ACE2 interaction is a well-established method which has been employed in many published papers, it is a minimum for the authors of this paper to provide the data from this simple analysis, regardless whether the RBD of RshSTT182 and RshSTT200 is able or not able to bind hACE2. The other functional assay is to use pseudovirus infection of hACE2 expressing cells. The authors need to provide data from at least one of the systems and better to have both

MINOR POINTS

- a) For Supplementary Fig 3, it is important to present a truly regional map including Japan and Taiwan, etc as *Rhinolophus* bats do exist in these places. Please also add in the distribution of *R. acuminatus*, the host of RacCS203 and *R. cornutus*, the host of Sc-o319.
- b) In Supplementary table 3, the % amino acid identity of the orf6 between RaTG13 ORF6 and SARS-CoV-2 is indicated at 1%?!

Reviewer #3:

Remarks to the Author:

This short article describes the detection of a SARS-CoV-2-related CoV in 2 of 5 R. shameli bat samples collected in 2010. Analysis of the near full-length genome sequence indicates that this virus is closer to SARS-CoV-2 in most part of genome than other SARS-CoV-2-related CoVs, except the receptor binding domain. Although the difference in receptor binding domain from SARS-CoV-2 spike, this virus maintains the highly conserved amino acid residues which are pivotal for SARS-CoV-2 spike and human ACE2 interaction. This study indicates that the SARS-CoV-2 related CoVs have much more geographic distribution at least in South Asian countries. This is an important work, but needs more data in terms of receptor analysis.

Comments:

1. Please provide data of ACE2 usage of the newly discovered virus. Because of difficulty of virus culture, authors can use spike and ACE2 binding assay, pseudovirus assay. ACE2 analyses from human and R. shameli are minimum.
2. Author please indicate sample type from which the positive CoVs were detected.
3. Page 3, 1 paragraph, line 8, "isolated" may be misleading. These two viruses were only sequenced in genomes, not cultured in vivo.
4. Figure 2B: Please label a and b in the figure.
Figure legend 2B: please indicate the regions or genes used in phylogenetic tree.

REVIEWER COMMENTS

Reviewer #1 (Remarks to the Author):

In the present study, Hul and colleagues described a novel SARS-CoV-2 related coronavirus in bats from Cambodia. This report is important and interesting and reads well.

We thank the reviewer for its positive and constructive comments.

A few concerns should be addressed.

1. Highlighting Guangxi and Guangdong in Figure 1b does not make sense because the pangolins were smuggled into the two regions from Southeast Asia.

We agree with the reviewer that the coloring of these regions on the map could be confusing, we have removed it.

2. A recent study described a SARS-CoV-2 related coronavirus from Thailand (<https://www.nature.com/articles/s41467-021-21240-1>). I suggest that the authors include this strain into their analysis.

We have added the virus described in this work in all our analyses, as well as another close relative of SARS-CoV-2 published during the revision period (<https://pubmed.ncbi.nlm.nih.gov/34147139/>).

3. The abstract section is quite descriptive and should describe more results about the two novel strains.

We have modified the abstract while keeping the word count below 150.

4. Interestingly, the two strains shared identical the same amino acid motifs critical for spike-ACE2 binding. It would be more convincing if the authors confirm the spike gene sequence using Sanger sequencing.

The spike gene of the two positive samples were sequenced using overlapping primer sets. The alignment of the sequences (~3900 bp) shows 100% similarity with the sequence determined by NGS (fasta file joined).

5. “than the any other closely related viruses discovered to date” Please remove ‘the’.

We have corrected this error.

Reviewer #2 (Remarks to the Author):

Hul et al., has retrospectively identified two SARS-CoV-2 related coronavirus genome sequences (RshSTT182 and RshSTT200) from *Rhinolophus shameli* bats sampled in Cambodia in 2010 which are genetically related to SARS-CoV-2. While the finding is not totally novel as similar viruses have been reported in bats from Japan, China to Thailand. The current findings further highlight that Southeast Asia is potentially a hotspot for SARS-CoV-2 related coronaviruses. The manuscript is concise and well-written.

We thank the reviewer for its constructive points.

However, a number of points need to be addressed to improve the manuscript:

MAJOR POINTS

1) If it is possible at all, the authors should include serology data (using archived samples is ok) to further support the prevalence of the viruses in *Rhinolophus shameli* bats in Cambodia

Unfortunately, we did not collect serum samples from these bats during the field mission in 2010 as it was difficult in small animals such as *Rhinolophus* and may result in animal suffering or death. We only collected oral and rectal swabs, and sometime feces when available.

2) For all of the molecular analyses, please repeat by including the recently published RacCS203 genome sequence (published by Wacharapluesadee et al in Nat Comm) from Thailand and the Sc-o319 from Japan

We have repeated our analyses with the RacCS203 genome, as well as the RpYN06 genome later published (the sequence from Japan was already included, but we used the "Rc_o319" name).

3) The ability of RBD to bind hACE2 is a hallmark of risk assessment for potential spillover of bat CoVs. The authors made some discussion/prediction in the paper, but purely based on protein sequence analysis only. As the biochemical analysis of RBD-ACE2 interaction is a well-established method which has been employed in many published papers, it is a minimum for the authors of this paper to provide the data from this simple analysis, regardless whether the RBD of RshSTT182 and RshSTT200 is able or not able to bind hACE2. The other functional assay is to use pseudovirus infection of hACE2 expressing cells. The authors need to provide data from at least one of the systems and better to have both

We have followed reviewer #2 and #3 recommendation, and have used a pseudovirus system described and shared on BEI resources by the Bloom lab. This new data reveals that pseudoviruses carrying the RshSTT200 spike are not able to enter cells expressing human ACE2. As control, we tested the ability of these pseudoviruses to enter cells expressing the ACE2 of *R. Shamelii* (RshACE2) (new panels in Figure 3).

MINOR POINTS

a) For Supplementary Fig 3, it is important to present a truly regional map including Japan and Taiwan, etc as *Rhinolophus* bats do exist in these places. Please also add in the distribution of *R. acuminatus*, the host of RacCS203 and *R. cornutus*, the host of Sc-o319.

We now include a truly regional map as suggested, and have added the distribution of *R. acuminatus*, *R. cornutus* and *R. pusillus*.

b) In Supplementary table 3, the % amino acid identity of the orf6 between RaTG13 ORF6 and SARS-CoV-2 is indicated at 1%?!

We have corrected this error, and added the values for the 2 additional genomes.

Reviewer #3 (Remarks to the Author):

This short article describes the detection of a SARS-CoV-2-related CoV in 2 of 5 *R. shamelii* bat samples collected in 2010. Analysis of the near full-length genome sequence indicates that this virus is closer to SARS-CoV-2 in most part of genome than other SARS-CoV-2-related CoVs, except the receptor binding domain. Although the difference in receptor binding domain from SARS-CoV-2 spike, this virus maintains the highly conserved amino acid residues which are pivotal for SARS-CoV-2 spike and human ACE2 interaction. This study indicates that the SARS-CoV-2 related CoVs have much more geographic distribution at least in South Asian countries. This is an important work, but needs more data in term of receptor analysis.

We thank the reviewer for his comments and suggestions.

Comments:

1. Please provide data of ACE2 usage of the newly discovered virus. Because of difficulty of virus culture, authors can use spike and ACE2 binding assay, pseudovirus assay. ACE2 analyses from human and R. shameli are minimum.

We have followed reviewer #2 and #3 recommendation, and have used a pseudovirus system described and shared on BEI resources by the Bloom lab. This new data reveals that pseudoviruses carrying the RshSTT200 spike are not able to enter cells expressing human ACE2. As controls, as suggested by Reviewer #3, we tested the ability of these pseudoviruses to enter cells expressing the ACE2 of R. Shameli (RshACE2). We also note that pseudoviruses carrying the SARS-CoV-2 spike are able to enter cells expressing RshACE2.

2. Author please indicate sample type from which the positive CoVs were detected.

We have added to the text the precision that all CoVs were detected from rectal swabs.

3. Page 3, 1 paragraph, line 8, "isolated" may be misleading. These two viruses were only sequenced in genomes, not cultured in vivo.

We have corrected the text.

4. Figure 2B: Please label a and b in the figure.

Figure legend 2B: please indicate the regions or genes used in phylogenetic tree.

We have completed the figure legend and added the labels.

Reviewers' Comments:

Reviewer #2:

Remarks to the Author:

The authors have addressed all of my concerns in the revised version.

Reviewer #3:

None